# “Returning to Ordinary Citizenship”: A Qualitative Study of Chinese PWUD’s Self-Management Strategies and Disengagement Model of Identity

**DOI:** 10.3390/bs12080258

**Published:** 2022-07-28

**Authors:** Apei Song, Zixi Liu

**Affiliations:** 1Faculty of Law and Justice, School of Law, Society, and Criminology, University of New South Wales (UNSW), Sydney, NSW 2052, Australia; songasa.711@gmail.com; 2School of Sociology and Anthropology, Xiamen University, Xiamen 610225, China

**Keywords:** self-management strategies, identity, PWUD, drug policy, disengagement model, police

## Abstract

How PWUD (people who use drugs) live under drug governance is an important research question. This study adopts a qualitative research method to explore how PWUD in China self-manage after perceiving the dilemma of incomplete citizenship and the social pressure brought by drug control arrangements. Through analysis of 130 PWUD’s files and in-depth interviews with 10 interviewees (from the 24 preliminary interviews), this study found that PWUD developed action strategies of hidden mobility (spatial isolation), disconnection of past experiences (time isolation), instrumental actions, as well as narrative strategies of reframing themselves as ordinary citizens with attempts of reversing identity disadvantages. Further, PWUD’s self-management strategies manifest as a disengagement model in which the actors (PWUD, not rehabilitation agencies) do not intend to develop integrative positive identities through dispersed, practiced behavioral strategies, but attempt to return to pre-addiction, non-socially exclusionary citizenship experiences. The disengagement model and its negative effect on PWUD’s social integration help us reflect on the current implementation of rehabilitation projects and institutional settings of drug governance.

## 1. Introduction

Social sciences often use substance dependence or addiction as an essential indicator to define the individual characteristics and identity situation of PWUD, to guide and establish various types of regulations and management to restrain and restrict them [1,2,3].

Thus, current management approaches and drug treatment regulations in various countries adhere to the underlying assumption that people who use substances such as heroin and methamphetamine (also known as psychoactive substances) develop intense, uncontrolled substance abuse and dependence, which leads to health risks and problems. Society has produced social control systems and management models for management and abstinence based on this general medical perception.

However, such studies only provide a range of social production after substance abuse has been defined (in medical discourse). e.g., the medicalized conceptualization of addiction and the development of rehabilitation treatment. However, they neglect and do not fully reveal the re-productive practices that occur during the development of social institutions (e.g., social integration in the post-addiction phase). PWUD, especially those in the detoxification phase, cannot simply be considered part of the “addiction-abstinence” process as individuals with mobility and social influence.

Beyond its relationship to material things, the act of PWUD is itself an extension of political interests and claims.

Achieving recovery and harm reduction are two important types of research that have been conducted around PWUD. The former combines the disciplinary products of psychology and medicine to unravel the physiological and emotional changes in addicts with substance abuse and addiction mechanisms and propose a pathway or framework for recovery treatment. The latter has a broader field of study and issues, focusing not only on how to achieve health medically but also on the health and dignity of communities, the implementation and enforcement of policies, and the negative impact of laws that produce [4,5,6].

However, refining these two types of issues presupposes the existence and living conditions of PWUD and requires real-time observation according to the changing times. However, such anthropological observation is not yet richly presented in the Chinese context due to the presence of social stigma and moral evaluation. In other words, the lack of description and analysis of the survival situation and living conditions of PWUD in China does not provide a concrete and realistic image that can be referred to in time for policy formulation, rehabilitation management, and social work programs.

This study illustrates how and why Chinese PWUD use self-management strategies to cope with the identity issues of drug governance and incomplete citizenship within institutional arrangements characterized by social exclusion. Empirically, this study focuses on PWUD’s daily life, specifically, the experiences and practices of de-stigmatizing life in the context of social control, which is the core of the study from a cultural and life-course perspective. The first part of the analysis describes PWUD’s lived experience of the identity conflicts and their perception of incomplete citizenship in the everyday life situation of drug governance. The second part examines their strategies of self-management to alleviate the identity conflicts and conceptualizes the model of disengagement. Finally, we discuss the extent to which self-management strategies and self-narratives of “possible citizenship” help them break through the identity predicament, gain life autonomy, and enhance social integration.

## 2. Literature Review: PWUD and Identity Research

There are mainly two types of issues on PWUD: how to achieve rehabilitation and how to reduce harm. The former focuses on the recovery of physical health to interpret the physical and emotional changes of PWUD through the mechanism of SUD (substance use disorder) and addiction [7,8,9,10,11,12]. The latter has a broader scope of discussion, following with rehabilitation treatment, primary physical health injuries, justice and recovery in social relations, political situation, and rights of PWUD [4,5,6]. Therefore, studies on harm reduction must pay attention to the influence of authority, family structure, and social structure on PWUD. It presents a top-down analysis context and relies on a politicized approach.

Using strategies to reduce harm is filled with the imagination of the governing subjects. In practicing these ideologies, some help the rehabilitation of PWUD, and some invisibly aggravate their harm, including social discrimination, damage to self-esteem, and leading an unhealthy life [13,14]. Lynda believes that understanding the drug problem requires value neutrality [15]. The government and service departments should be responsible for their remarks to guide the implementation of treatment as soon as possible. 

The ideology of harm reduction guides the research and practice of social identity rehabilitation of PWUD to assist them in achieving physical rehabilitation and reconstructing their identity in society. These studies on reconstructing the rehabilitation social identity of PWUD are mainly structured [16,17,18,19], using structural power to assist PWUD in achieving rehabilitation. Structural power presupposes the type and situation of PWUD and sets up what they should look like after successful rehabilitation. It is hoped that addicts will no longer be shackled by addiction in the future. In his discussion of the battle for chastity, Foucault sees Gesian’s steps in outlining the struggle for chastity as a process of continuous abstinence, intending to allow the subject to achieve “unmoved by evil”. Reconstructing the identity also aims at rehabilitation and abstinence, expecting that the addict will not be shackled by addiction in the future. Its operation can be broadly summarized into two categories: structural and critical shaping.

Structural shaping refers to constructing normative identity acquisition pathways through structural forces, mainly including “state–citizen” and “social network–citizen”. For example, drug research in Australia has focused on how to solve drug use problems and effectively change the situation of PWUD from the perspective of law and drug policy, tensions, and dilemmas [1,16,17,20]. However, this shaping, with arrest, incarceration, and reform through labor as actions, reinforces PWUD’s fear of criminal justice, and although detention is always in the name of detoxification, it creates persistent occupational hazards and integration dilemmas in practice [21,22]. Therefore, PWUD either use Alice Goffman’s proposed escape route to evade capture or actively resists criminal justice to build the moral system of the incarcerated community in the form of criminal “honor” [18,23]. 

In addition to policy and law, governmentality also plays a vital role. Beth proposes neoliberal governance techniques that focus on how governments intervene with populations to (re)produce ideal citizens. For example, the development of biopolitics focuses on a biological perspective on population, with surveillance and individual responsibility pressures as strategies to preserve population health. Thus, they emphasize the critical relationship and combination between biological identity. Additionally, the state, the market, and biomedicine focus on reparations for collective personal harm as the possible governance models for biopolitics. Scholars such as Ingrid discuss how public health strategies can reshape forms of autonomy and citizenship for people living with HIV, and studies have found that exemplary citizenship for stigmatized groups is difficult to follow through and sustain because of the complexity and inequality [24].

The “social network–citizen” manifests itself as a process of shaping PWUD by relational forces. Family and community are among the essential spaces for identity remodeling and social identity reconstruction, assisting PWUD in transitioning from social networks of abuse to recovery-supportive social networks [25]. Especially the process of identity change within recovery is essential, through which new identities replace stigma and addict identity, and recovery capital and quality of life are improved [19].

Critical shaping refers to the critique of the established set of criteria that define drug users as a stigmatized, marginalized, and pathologized group, manifested in a critique of medical and biological discourses and an emphasis on neoliberal self-management. In classical studies, PWUD’s bodies are constantly reduced to tissues, blood, and neural pathways as a de-subjectified generation of enjoyment or pain [26]. To argue and understand how the body is manipulated by the “substance” and whether it is accepted or feared in response to the manipulation, thus constructing a negative association between certain addictive behaviors, especially drug use, and physical, psychological, and neuronal [27,28,29].

Medical discourse has effectively promoted the healthy expression of the drug phenomenon and the pathologization of PWUD [30]. The eugenic philosophy of biological development in the first half of the 20th century gradually popularized the conceptual model of non-defective joint disease, making PWUD biologically judged as vulnerable, defective, and dangerous individuals. Thus, it critiqued the medical and biological model as an essential research direction since the late 20th century [31]. Two of these critical assumptions are relatively well established.

The first promotes de-pathologization and de-biological differentiation, thinking about drug addiction on a relational level [32] and from a cultural perspective [26]; the second is eclectic, using a balanced or integrated strategy outside of material determinism (neuroscience) and cultural influences [33,34], focusing on the narratives of drug users and the complexity and diversity of PWUD’s actions in social settings [35,36]. In narrative studies of PWUD, positive risk-taking and self-management characteristics are evident. For example, in the Swiss “recreational” using narrative, PWUD reflect on the blurred division between “safe” and “dangerous” and advocate for a change in the old judgments of “drug use” in the addiction recovery space to re-examine the association between drug users and drug use behaviors [35].

There is a clear difference between structural and critical shaping: the former takes the medical–biological discourse as the basis, through the power of the state and the power of relationships, to enable drug users to reach a standard of “health” in a normative and legal path. The latter critiques medical and biological discourses as being too homogeneous in their means, advocates attention to PWUD’s plurality and complexity in social contexts by reconstructing authoritative discourses and balancing cultural–biological positions, and emphasizes understanding PWUD’s narratives to help them develop self-management skills. The significant commonality between these two shaping approaches is that they both aspire to adjust the relationship between the disordered “individual” and society so that they can live without discrimination and exclusion.

Some problems cannot be realized in transforming the value of harm reduction from ideology to practical operation. Analyzing this gap, we need to pay attention to how the rehabilitation process of PWUD works in specific situations. The real problems are different from the original intention of reducing harm. The initial purpose of constructing identity is perhaps to reduce harm, but in the practice of management, if constructing ideas that ignore the opinions of PWUD, this political system creates all kinds of contradictions in grassroots practice. Therefore, it is necessary to carefully examine the understanding and actions of PWUD, take actions to deal with governance and difficulties, and reflect on the limitations of political imagination in governance.

## 3. Method and Analysis Framework

### 3.1. Field and Field Work 

The research was conducted from November 2019 to July 2020, cooperating with Social Organization Q, a police station, NCO (Narcotics Control Office), and three communities. The whole process of data collection and analysis can be interpreted as three successive stages: field work, in-depth interviews, and data processing. In the first stage, we entered the field and developed cooperation with relevant institutions, and gained access to PWUD’s profiles and project participants, which made in-depth interviews possible. In the second stage, we designed our interview outline and snowballed eligible interviewees with the principle of saturation. In the last stage, we translated, verified, and coded interview data through open and spindle coding, which laid the methodological foundation of empirical findings about incomplete citizenship and self-management strategies. 

Social Organization Q is the gatekeeper and main introducer of our fieldwork, through which we developed connections with the local police station, CAG (community assistance group), and three communities willing to introduce their drug rehabilitation projects. In addition, we selected PWUD interviewees from Q’s registration system that covered drug rehabilitation project sites across five blocks and recorded more than 130 biographical profiles of registered PWUD (66 males, around 73%, and 24 females). We asked Q to give referrals at the beginning of fieldwork and snowballed the following interviewees through them to provide other survey objects belonging to the overall research target and selecting the following survey objects according to the former’s recommendation.

### 3.2. In-Depth Interviews and Interviewee Selection 

The materials of the literature research include drug rehabilitation files and legal policy documents. Q provides detailed documents for PWUD. We also collected and analyzed related policies and legal documents from official websites, including the Drug Law (2008) and the Drug Rehabilitation Regulations (2011).

The in-depth interview was divided into two parts: an interview with PWUD and with staff. The interviews with PWUD focused on the following aspects: perception and response to stigma, understanding and narrative of drug use, health significance, national drug control agents, strategies, and actions integrated into life (Appendix A). Then we talked with the NCO and CAG in four aspects: management methods and requirements, how to carry out the agency’s tasks, how to distinguish and judge PWUD’s rehab status, and how to interact with them in the agency. After finishing the preliminary interviews, 14 PWUD and 10 staff were interviewed. We stopped here because ten in-depth interviews reached saturation. Then, we focused on the follow-up interviews with 10 participants whose narratives constitute the core points of this article. The specific situation is shown in Table 1.

Among the nine participants in the follow-up interview in Table 1, the top seven are PWUD, among which Ping (D), Wen (F), and Zhong (G) are women. Song (C) has the highest education and graduated from university. The others generally graduated from junior high school or elementary school. Respondents have a history of taking drugs for 5–22 years, and their drug consumption mainly includes heroin and synthetic opioids (methamphetamine). These participants were detained at least once in a detention center or a CDI (compulsory drug institution) for drug-related crimes. Xin (B) is currently in a relapse lawsuit, but it is challenging to be sentenced to a CDI for physical disease. The seven PWUD are all in the community rehabilitation stage. The CAG needs to regularly pass urine tests and interviews to monitor whether they are eligible for relapse or stable life. The eighth participant is the social worker responsible for communication in the CAG. He has worked in the project team for one year and six months and is familiar with various tasks and each registered PWUD’s personality and attitude. The ninth participant is the director of the NCO. He is very clear about the diverse work of drug control. During the interview, he cited drug policies proficiently and described the operation of grassroots public security organs (police stations), community assistance, education, and social organizations. PH is a local policeman, mainly in charge of drug rehabilitation and anti-drug work in the community. He has been a front-line policeman engaged in drug tasks for many years. We recorded and sorted 200,000 words of interview materials and a total of 50,000 words in various archive materials, regulations, and publicity manuals.

However, since the article is composed of 10 participants with different identities and experiences, it inevitably focuses more on understanding the management style and behavioral possibilities of PWUD, unable to provide general conclusions and grasp the overall drug governance features.

### 3.3. Data Processing and Coding

Firstly, we transcribed and verified the collected data. In the process of transcribing the audio recordings, we also blurred the anonymity of the interviewer and the prominent third-party information and address information, and during the verification process, we verified the transcribed material word by word with the information from the audio recordings, marked the emotions of the interviewer in the audio recordings, and added some unspoken information through context. The process of translation and verification is shown in Appendix B
Table A1. 

Secondly, after completing the verification of the interview materials, we conducted a coding process of “conceptualization-generalization-identification of core genera-analysis framework and patterns”, which was divided into two coding stages: the first stage was the open coding stage, which mainly decomposed, examined, compared, conceptualized, and generalized the interview materials, including labeling, genera, attributes, and dimensions (see Appendix B Table A2); the second stage was the spindle coding, which regrouped the genera and identified core genera according to the framework of “context-restriction-behavior-result”.

Based on the primary coding, the researcher followed the coding logic process and performed secondary spindle coding. In Table A2 we find the existence of restrictions and supervision of PWUD’s social moving, so how did they deal with this situation? We explored this issue through a deeper generic analysis of “context-restriction-behavior-result” modeling (see Appendix B Table A3)

After organizing, merging, and cleaning up the relevant interview materials through two coding sessions, we were able to break down more clearly the core questions about how PWUD live under governance and how they manage themselves: (1) How do PWUD understand their position in governance? (2) How do recovered PWUD run their own lives to reduce the constraints imposed by governance and identity? (3) How can we understand the range of behaviors and perceptions of self-management of PWUD? Can they be helped to change current limitations?

### 3.4. Ethical Considerations 

The research is supported by the Department of Sociology at X University and Q Organization. Before starting the interview, I explained to the participants that no private information would be involved. The materials obtained are only used for research, and all participants are anonymously protected during the investigation. Each participant has the right to refuse to answer sensitive questions, and the researcher could not force the expression. They also can withdraw from the interview at any time, and they can say no. 

## 4. Effect Results of Multiple Governances: Incomplete Citizenship

The historian Feng believed that it is imperative to pay attention to the diversity of PWUD’s identities in the process of drug control [37]. A series of anti-drug campaigns in China (In 1729, Emperor Yongzheng issued the first ban on drugs. From the beginning, the Qing government insisted on its political stance against drug smuggling. At that time, the elite class believed that “opium (consumption) can alleviate people’s sense of loss feeling”(Boliwei, 2017:270). At the beginning of the 20th century, China’s rising nationalism provoked controversy about opium consumption. Opium was regarded as a national humiliation and was one of the main reasons for declining China’s national power. There were three anti-drug campaigns, which are the 10-year anti-drug plan in 1906, the six-year anti-drug plan in 1934–1940, and the new China’s anti-drug campaign) which made PWUD not only consumers but also victims and lawbreakers [38]. These multiple political identities have continued nowadays. Moreover, in the past research by authors, we showed the multiple governances here [39].

NCO is relatively mediocre and is mainly responsible for drug control and crime-fighting. It tends to treat PWUD as prospective citizens, and its main work philosophy is to guide all citizens to participate in drug rehabilitation and improve the transformation mechanism for prospective citizens. On the one hand, it promotes and educates drug knowledge and trains ordinary citizens to realize the harm of drugs and consciously manage physical health; on the other hand, in drug rehabilitation, NCO urges PWUD to get rid of addiction through risk ratings.

The primary jurisdiction of the police station is to control and prevent PWUD’s relapse and risky behaviors through urinalysis, hair testing, and unannounced visits. The central work concept is to maintain “society safe”, which means controlling them in the jurisdiction to avoid spreading drugs and other illegal activities. Therefore, PWUD are regarded as potential risk-takers of public security.

The CAG needs to conduct micro-day-to-day interactions and integrate assistance, education, and supervision, although the CAG also collects some needs from PWUD and provides complementary services, such as docking vocational training, temporary work resource links, etc. The task of notifying the urine test once a month obscures the unique nature of the help, support, and education of the CAG team. Instead, it also emphasizes that CAG members are alert to the risky behaviors of PWUD. PWUD are regarded as a group that needs daily supervision at this level, not the object of being helped and taught.

As a result, different executive bodies (NCO, police station, CAG) have formed multiple governance practices in drug rehabilitation management. Additionally, the multiple governances in drug rehabilitation have created the identity dilemma of the PWUD in action—multiple and incomplete identities. Multiplicity is manifested in that the national drug policy, law, and management departments have given PWUD multiple and contradictory identities. This split identity makes it impossible for users to rely entirely on laws and policies to implement their obligations and rights and causes difficulty maintaining identity recognition in each system stage. Among them, the differences within the agency department need to be more cautious. It can be expected that the identification of patients or offenders/deviants cannot inherit and implement ordinary citizens’ screening functions, nor can it subtly change the definition of criminals stigmatizing “substance users” by the public [40] (see Table 2).

Incompleteness is manifested in two aspects: one is the limited acceptance by the public of the civic status of PWUD. Drugs are closely connected with national security in modern Chinese history. The danger of drugs has been continuously strengthened and shaped in the story of modern colonial history. In the author’s investigation, PWUD cannot be called “patients” or “deviant”, nor even lawbreakers in the general sense, but “criminals” signals. This frequently appearing label does not come from the law, but it flashes in the community life and cultural life. The “drug use history” has become a sharp edge in separating personal social relations.

Second, incompleteness is also manifested in limited acceptance in professional employment. During drug rehabilitation, PWUD are mainly engaged in some part-time jobs. During these three years, the local grassroots public security departments are not given a non-criminal certificate, so PWUD take on higher occupational risk when facing work change or economic pressure.

Structural deficits also prevent the vision of a unified identity. National drug laws and policies and the agency sector have given multiple identities to PWUD to facilitate administration. Its fragmented identity prevents PWUD from relying entirely on drug laws and policies to become legitimate citizens again and from maintaining a perception of sameness at different stages of the drug treatment system. Among these, the differences within the agency sector require even more caution. It is possible to anticipate the identification of patients, law and order disruptors, or daily supervision groups [39], the inability to inherit and implement legally predetermined citizenship programs and functions, and the inability to subliminally change society at large’s stigmatized definition of drug addicts as criminals.

The structural level reveals the lack of an institution in the current anti-drug system that connects the rehabilitation center’s macro and micro levels. Rehabilitation centers generally exist primarily based on the market and can provide a richer, more diverse force for drug treatment participation beyond political forces [41]. The CAG attempts to play a similar role to some extent. Still, this organization has no actual workplace or space and is often present and working at the same time as community workers. PWUD are unaware that the CAG is separate and dedicated to their sector. In addition, rehabilitation center staff need to have specialized experience in addiction treatment to ensure that they can effectively play a rehabilitative training and treatment role. They effectively and more professionally provide services that can replace the “half-assed” CAG in drug rehabilitation. On the one hand, it would help provide a particular and more trusting space for PWUD. On the other hand, it can clarify the responsibilities of the community help groups to avoid the constant fragmentation of the organization’s sense of value by combining rehabilitation services and supervision and control work.

There is a lack of rehabilitation centers and a lack of systematic room to operate on how to arrange and dispose of PWUD who participate in voluntary drug rehabilitation. The experience of Ma, the head of an anti-drug social organization, provides an example of a reasonable attempt in the city of Chengdu in 2020 and highlights a structural problem: the lack of institutions dedicated to voluntary drug rehabilitation. Some of the institutions labeled as “voluntary drug rehabilitation” are set up within and are subordinate to CDI, preventing them from being truly voluntary. As a result, in the dilemma of “incomplete citizenship” for PWUD, how can the process of identity reshaping be achieved after unsuccessful appeals to the external social environment? Thus, exploring the process of building a self-contained and defined self for PWUD requires concern for individual conflict management strategies. The emergent self-management strategies of recovering will help interpret how the multiple structural governance differences can be reconciled and reconciled in action and how the issue of incomplete citizenship can be changed and resolved.

## 5. PWUD’s Incomplete Citizenship and Self-Management Strategies 

### 5.1. Perception of Incomplete Citizenship in PWUD

When shifting from the perspective of structural analysis to the discussion of actors, the most important thing is to understand the cognitive differences between the role of the analyst and the addicts themselves. Lan said, “Under the long-term acceptance of state agency authoritative leadership and governance, I inevitably from an epistemological perspective and interactive mode of observation and understanding” [42]. We need to change our position from a speculator to a drug addict perspective.

Discussing the drug social plan and rehabilitation management methods from a structural perspective explains that the identity dilemma that has been produced from the establishment of the government anti-drug project to the continuous development (time dimension, local difference) is not conducive to the rehabilitation and re-citizenship of PWUD. However, actors who have been in the grassroots drug rehabilitation stage for a long time do not have historical and expansive perspectives and think about the identity discrimination and difficult life they have suffered. When they perceive and understand their situation, they will base it on reality. Specific communication subjects in life, or stress events, make simple judgments and general impressions. What dominates them to form this relatively narrow cognition is often the sudden strengthening of emotions at a specific moment.

PWUD’s limited perception of their hardship is mainly aimed at a few specific subjects with a high frequency of contact in daily life: police, social workers, community workers, and other citizens. The interpretation of these role management methods in the previous article conforms to the image of PWUD in their lives. Therefore, from the perspective of the trust relationship, the trust of PWUD is roughly presented as: “NFO > CAG > police station” of course, this kind of trust performance is not a fixed pattern. The occurrence of conflicts and strong emotional expressions will always affect the trust situation. PWUD develop a flow of cognition and trust in other subjects, forming their interpretation of their identity dilemma. However, the discomfort of fragmented and different identities is challenging to coordinate and keep self-consistent.

### 5.2. Three Self-Management Strategies 

PWUD must cope with the police law enforcement, frequent urine examinations, and daily supervision of the CAG in everyday life and avoid the repeated expression and memory reappearance of drug use experience and avoid the obstacles and adverse effects of drug rehabilitation on life. Combining their limited resources, PWUD have developed unique identity management strategies from the perspective of social space, time, and resources, to change the dilemma of “incomplete citizenship” and develop a more “dignified” and “resourced” social life.

#### 5.2.1. Spatial Isolation: Fleeing from One Another in Urban Communities

The Drug Law and the Regulations stipulate PWUD’s mobility during detoxification. If they leave the city (where they registered), they need to use their ID cards to “register” in the community. Only then will they be allowed to take long-distance transport and book hotels. The digital information network unreservedly shows whether PWUD are in the detoxification period.


*When I went to Fuzhou to visit relatives, I had to register the type of case, the reform through labor, the handling unit, and family information. Who is my wife? My phone call number? Although I was issued this certificate, the local police station took me for a urine test the first day I stayed in the hotel. (Feng-interviewee A, 2019)*


Internal-city mobility is often unrestricted. However, there is a risk of personal information exposure compared with the restrictions on outbound movement. Even in an interview with Huang, she mentioned that she had only recently obtained the contact information of a PWUD who had been in the CAG for two years and learned that this user had other residences.


*Now there are only two people in my family, me and my father, and everyone else goes to seek refuge with my uncle. Because what I did before was so disappointing, my uncle lost trust in me, and now I am not allowed to seek refuge... (community). They call me once a month. I will not force an interview on the phone. Before, I lived with my girlfriend in Nanping city (Min-interviewee E, 2019).*



*I worked in Zhangzhou city before, and the community didn’t care about it... This time I came out of the CDI. I might not go home and live with my aunt. My aunt’s son is still willing to take care of me (Song-interviewee C, 2019).*


Users show two modes of hidden movement in the city: one is the “smuggling” mode. Take Min as a representative, he secretly left the registered address, did not take the initiative to inform the community whereabouts, and some in the community inquired about his specific situation. Communication targets of the CAG are only achieved through quick telephone contact. Next, the “exile” model is represented by Song, who did not mind the community asking about her current address because she had many mobile places. The CAG also cannot know the movement and residential address.

These two modes of concealment and mobility often do not conflict. The “smuggling” mode focuses on the concealment of the outward destination. It is easy to fabricate the help and education group’s requirements if it is a short trip. The “exile” mode emphasizes the concealment of real daily life places to avoid sudden police random urine tests or door-to-door visits with members of the CAG. They (CAG and the police) have caused damage to personal and neighbor relationships because they will be exposed to their neighbors as PWUD. The neighbors would get very tired of them. Additionally, these two modes can be flexibly switched at any time depending on the action content of the CAG.

The two models above reflect a common approach to identity conflict: the concealment of stigmatized or marginalized identities is achieved through hidden flows in space. They are also separated from the accessible surveillance space of the CAG and police to a certain extent. Space isolation makes the identity defined by the CAG and police for users stuck in the community, and completely unfamiliar social relationships and impressions in the new space can be reshaped and presented. It is a very radical way of handling identity to a certain extent. The radical manifestations are sudden changes in space or a specific intended change in stigmatized identities and re-operation. It is also a processing method that requires great courage.

It is necessary to abandon the original living space’s resources and relationships. It is also essential to carefully maintain the new space’s identity from being exposed to digital and information networks. When Min lived in his uncle’s house to start a new life, his uncle was disgusted with him because of improper management, repeated relapses, and theft. Re-operating the legal identity image is the goal of spatial isolation. Once it fails, it can only be re-operated in another area.

#### 5.2.2. Broke: Abandon the Past and Look Forward to Relieving 

In addition to managing identity conflicts in a spatially isolated manner, the participants also presented a way of mitigating identity conflicts at the temporal level, striving to separate themselves from past using drug behaviors, and shaping a new self.


*Now I have no other plans. When the community rehabilitation is over, I will take another driver’s license test, run a taxi, or be a Didi (private taxi company) driver... I live with my parents, and their pensions can support me partially. Don’t worry about finding a job. Drug use ruined me. I used to be a bank clerk. It is impossible to go back to my life in the future, but if I want to start again, I can’t get drugs (Song-interviewee C, 2020).*



*I don’t want the police to come to my house suddenly to take urine samples. I have tried very hard to control and manage myself to remind myself of my past using a drug and cherish the current life of family reunions. Still, every urine test reminds me of my past using drug behaviors. I can’t do it without community rehabilitation. Let yourself start again (Wen-interviewee F, 2019).*


Respondents in the narrative who showed a strong tendency to separate from past drug use behaviors are looking forward to the coming of relieving the CAG and finishing drug rehabilitation. They imagine that they will seal the past bad behavior from the time dimension and start a new life without these dynamic controls. Both Song and Wen hope that using the drug is just an experience, which means the using experience has little influence on their everyday life and relationships with their girlfriends and children.

Every contact of the CAG invisibly reminds them of various connections with past using drug experiences. The sudden urine test forced Wen to recall her past drug use experience and made Song interrupt his living arrangements. Thus, the explanation does not end, and isolation from the past self is an imagination that is difficult to realize fully. The current break with past behavior is only an exploratory effort.


*There is no way...I also thought about driving Didi and taxis before. If you have experience in drug use, you will find out they don’t provide the chance for you. Except for part-time jobs in society, which one is better? Will not open the door for PWUD (Feng-interviewee A, 2019)*


Feng presented his own experience. Isolation is only a temporary conservative strategy, and social integration is not as simple as imagined. The public’s understanding of the identity of PWUD will still lead to job discrimination. 

Separation from past behavior is a virtual action, but it is unrealistic for individuals to want to ignore activities they have done in the past. Past life experience has been internalized in the individual’s cognition and behavior habits. Isolating the past, looking forward to the future, and preserving the present can only be an individual’s inner choice and attitude and cannot face the unity of the past, present, and future. They have never been able to meet the dilemma of social integration and multiple identities, nor can they reconcile with their past self. Of course, this cognitive strategy of separating the past has relieved the pressure of identity dilemmas to a certain extent and provided the addicts with the motivation and inner strength to continue to persist.

#### 5.2.3. Instrumental Cooperation: Using Identity to Obtain Resources

Compared with the previous two strategies, the separation and concealment of stigma in space and time, instrumental cooperation has limited the spread of identity conflicts and focused on using identity cognition in different agents to achieve and realize beneficial interests through formal collaboration. Specifically, it can be manifested in three cases: receiving various subsidies, issuing a non-crime certificate, and cooperating with the police station to exchange urine tests for resources, like Min and Ping:


*2008/5/22 sensitive glaucoma treatment application assistance; 2009/4/21 want to participate in methadone treatment; 2009/6/15 asked if I can help apply for the cost of methadone replacement treatment; 2009/10/14 take methadone invoice to apply for subsidies; 2009/11/16 Take the methadone invoice to apply for subsidies; 2015/7/13 Min wants to apply for subsistence allowance again; 2018/7/25 Min hopes to apply for affordable housing; 2019/4/26 Min currently uses for subsistence allowance as a family household. Submit the approval materials; 2019/5/9 Min’s family subsistence allowance is approved; the 2019/12/19 roll to the security room number can get a two-bedroom and one-living house (Min’s file).*



*I had received a community subsidy for breast cancer before, and my brother (Min) also got money to treat glaucoma. The community asked me, the Guangming Fund, to reimburse the cost. Later, we went to the community if we had a problem. They are more reliable. The sewer was blocked last week, and I also asked the city to solve it (Ping-interviewee D, 2020).*


Since 2008, Min has realized that drug use experience and offenders’ and patients’ identities can help him obtain various government and social resources linked to the CAG more easily. The most prominent feature of these resources is that Min is not clear and difficult to access. Min only needs to show a firm attitude towards drug treatment during the whole process. Then he will give difficulties and needs a priority. For example, the affordable housing policy is a policy for urban residents with low incomes who cannot afford to buy a house. However, users are more likely to be packaged into low-level people who urgently need affordable housing. Therefore, Min and Ping applied for the minimum guarantee fund faster, applied for affordable housing in December, and got a two-bedroom and one-living room.


*I remember that Policeman Chen from the police station gave Liu, who had already been expatriated, a non-criminal record and could open it. Liu’s son served as a soldier, and his family needed certification to prove his innocent background. Liu came to the police station to ask Chen, and Chen gave a non-criminal certificate before Liu’s son could join the army. Chen regretted it. All ex-posters can open a non-criminal record or certification, but the risks need to be borne by policeman Chen himself, and the pressure is immense (Huang-interviewee H, 2019).*


In another case, to help his children serve as soldiers, Liu went to the police station and asked for a non-criminal certificate, proving that Liu’s family identity is innocent to the government auditors. 

A non-criminal certificate is a kind of certification material to show the legalization of citizenship. It also means that users should complete community drug rehabilitation, and detoxification will not leave a record in the public security organs, which further proves that the identity of criminals considered by society is misunderstood and distorted. However, the general security organs are unwilling to issue a non-criminal certificate for PWUD. After all, users are different from ordinary law-abiding citizens. They are still a factor of insecurity in real society. If a non-criminal certificate is issued, the local public security agency must guarantee that the person will not violate the law or discipline. However, as far as the general security agency is concerned, its fundamental responsibility is to ensure local security. Various task indicators do not indicate whether it is feasible to issue a non-criminal certificate. That makes the police themselves bear the later risks of political behavior and rules, which police officers regret.

Xin, nearly 60 years old, has a severe illness and can only detoxify in the community. During the 2020 epidemic, he lost contact for almost three months. Accounting for Xin’s past using drug history, the CAG believes that he has relapsed. However, in May, Xin took the initiative to contact the community anti-drug committee to indicate that he was willing to undergo a urine test, showing that he could not be arrested immediately. The community must assist in applying for a retirement pension. Even more paradoxical is that the community and the police agreed to his request because of bureaucratic management’s complexity. If Xin is caught by another police stations, the paperwork and handover work between them and other agencies will be very tedious and troublesome. To avoid extra work brought by the bureaucracy, the local community and the police station agreed to Xin’s terms. At the end of May, the CAG accompanied him for a urine test. When leaving the police station, I observed:


*Xin was still a little rickety, turning to the alley, rubbing his hands slightly faster, his face calm. A middle-aged man in white is standing nearby smoking a cigarette. “Xin?! Why are you here?” “...we are...waiting for someone” Xin’s voice was calm. “Didn’t you go to take a picture?” “I’m from the police station” The middle-aged man in white, throwing away his cigarette butt, immediately stopped our inquiry.” Then let’s go first.” The anti-drug station workers continued to walk and “leave here first” I couldn’t help but feel the urge to run away. When I came to the alley, I turned my head back, and a group of people gathered in front of Xin like a swarm of people (Observing at the police station, 2020.5)*



*“They have to make up the capture process. If other police stations catch them, they still need to make up various procedures” (Huang-interviewee H, 2019)*


Xin’s interaction with the police station is an exchange logic. He cooperated with the police to complete the arrest in exchange for promises such as urine testing done by the law (This point of view is emphasized in every anti-drug work instruction from 2014 to 2019. For example, the theme in 2014 is “Persevere in carrying out the anti-drug work in-depth”, the article in 2015 is “Unswervingly Win the People’s War Against Drugs”. The theme in 2018 is “Taking the path of governance of the drug problem with Chinese characteristics and resolutely winning the new era”. There was also the “People’s War Against Drugs” and “Never Retreat Without a Complete Victory”, in 2019), letting him go home that day, and the community as soon as possible will assist him in applying for a pension. Further study reveals that this real-time interactive performance is based on the comprehensive understanding of both parties’ identities for drug use. Xin continued to talk about the dilemma of his identity as a patient and a lawbreaker, prompting the police station to be unable to enter the camp of severe illnesses. The police station understood that Xin’s age and physical condition would not negatively impact his social impact even if he relapsed, so he asked Xin to complete the arrest plan and case filing.

Instrumental cooperation does not mean acceptance or active integration. It is not about accommodating multiple identities into a single positive identity but strategically using the advantages and disadvantages of different identities in interacting with the CAG and social citizens to present a more suitable identity. As a result, instrumentality focuses more on the assembly of multiple identities to reflect conflict and exert creativity.

The most prominent commonality of the three strategies for managing identity conflicts is that drug problem management techniques have been transferred to individuals who demand the freedom to carry out their lives [16]. PWUD need to self-determine and manage identity conflicts and risks. They face comprehensive and complex multiple identities compared to the NCO and CAG. Multiple identities conflict inevitably maintains a high relapse rate, and the government’s security and crime control goals have to pay a higher political price [43]. 

For this reason, the strategies of concealed mobility of PWUD (spatial isolation), disconnection from the past (temporal isolation), and instrumental use of identity can help enrich the information at the actor level under the operation of political behavior and show the process and purpose of self-management that individuals reassemble multiple identities and creativity. The drug rehabilitation policy is a goal-oriented ideal product of dominating the culture and absorbing the improper operation of political behavior from marginal groups’ real plight and life, thereby making up for the coexistence of multiple logics of the law and multiple inconsistent governances.

More importantly, these three differentiating strategies are not only simply classified as ideal types. In being narrated and performed recurrently, PWUD have been repressed and constrained for a long time, constantly and actively re-adjusting and sticking them together. Depoliticized expression is the characteristic of this process rather than the political characteristics of structural shaping. 

### 5.3. Possible Citizens: The Purpose of the Self-Management

The relationship between self-management strategies and incomplete identities is not a correspondent correlation. The most striking commonality between the three strategies for managing identity is that the drug problem techniques have been transferred to individuals who demand the freedom to conduct their lives [44]. PWUD need to self-determine, perceive, and manage identity conflicts and risks. Different institutions such as the NCO, CAG, police stations, and CDI require PWUD to face an integrated, complex, and fragmented identity. The incomplete citizenship inevitably sustains a certain high rate of relapse, and the government’s security and crime control goals are consequently more politically costly [43].

In Figure 1, the strategies of hidden mobility (spatial isolation), disconnection from the past (temporal isolation), and instrumental use present the process and purpose of individual reassembly of incomplete issues and creative self-management. These strategies require that drug policies need to draw from the real dilemmas and lives of marginalized groups, thus compensating for the dilemma of the coexistence of multiple logics of law and the plurality but inconsistency of governance.

More importantly, these three differential strategies are not only ideal types that are simply delineated; in their being narrated and exhibited repeatedly, the long-suppressed and cramped selves of the PWUD are constantly and actively re-adjusted and glued together. Just as Fitzgerald’s book has seen life as a process of collapse, it constantly undergoes blows that eventually make it self-collapse and shatter (see Fitzgerald F.S. (1945). “The Crack-up”, Chinese Translation Version, 139–145 [45]). 

The space of order in which the PWUD lives reproduces the higher frequency of the “blow”. However, as Fitzgerald affirms in the individual’s ability to adjust and reshape himself, even those who have been rejected and humiliated still have the possibility to bind the fragments and prevent collapse. In the lives of PWUD, this is expressed in the specific binding of the self through “hidden mobility (spatial isolation)”, “disconnection from past experiences (temporal isolation)”, and “instrumental strategies”.

The identity management strategies of PWUD are dedicated to the “possible citizenship” of narrative becoming to cope with the identity problem of incomplete citizenship. Specifically, presenting themselves as having the ability to be ordinary citizens, manifested in the attempt to cope with identity issues through actionable strategies to hide their personal experiences of using drugs, carry out social and work routines, and run their social lives like ordinary citizens.

They express their support for national guiding policies and top leadership, recounting themselves as part of good citizens who love the country and support the president. For example:


*“They (some other countries) are not taking the COIVD-19 seriously. China is better at doing prevention and control. Foreign people support the pursuit of freedom, they are in pursuit of those, you could ask them, can do as we do? I don’t even know......” (Wen- interviewee F, 2020-11)*



*“(Military program review) our country is strong, (referring to military exercises) the territory of Nansha cannot be occupied, I support such exercises, we are also a part of the patriotic......” (Song-interviewee C, 2020-10)*



*(President Xi speaks) Xi is thinking for us, we also want to engage in building (the country), and no one wants us.” (Zhong-interviewee G, 2020-10)*


PUWD constantly reiterate that, apart from their past experiences with drugs, they are more supportive than others of the government’s and the country’s big decisions and more willing to defend the rights and interests of the territory and the nation. The high degree of consistency in the “right and wrong” attitude emphasizes that they share the same position as other citizens. The individualized actions of spatial segregation, disconnection of past experiences, and instrumental strategies all work to achieve or assist addicts in their quest for an ordinary, dignified, and independent (non-police-dominated) life.

This narrative of “possible citizenship” was developed by individuals as a way and pathway to ordinary citizenship, primarily to replace drug policies and legal assumptions that were not realized after implementation by agents at all levels. Interestingly, the interviewees did not truly understand and clarify the content of the laws and policies relevant to their group, and their sense of conflicting identities stemmed primarily from the demands of police and CAG support teams to control, monitor, and differentiate them from ordinary citizens. Although PUWD do not truly understand the various aspects of their difficult situation, this does not affect their admiration for non-stigmatized identities and mainstream groups.

The narrative and self-management strategies of “possible citizens” have political legitimacy. For PWUD, they gradually realized that the cause of their situation was the society’s model of inclusiveness and social control and not just their problems, such as their past addictive behaviors. It strengthens the determination of the PWUD to self-manage and even gradually develop a systematic pattern and logic of behavior.

## 6. PWUD’s Identity Disengagement Model

The structural-level management approach creates layers of tightly controlled existential space for PWUD, which invariably affects the lived experience of drug users [39], especially after feeling the identity issue and bankrupt citizenship program, they adopt strategies such as hidden mobility (spatial isolation), disconnection of experience (temporal isolation), and instrumental strategies to hide the personal experience of drug use and try to return to ordinary citizenship. This personally developed model of citizenship programs enacts the entanglement and tension between experiences of drug use, self-management, and political models. After perceiving the pressure and cramp of the structural situation, the identity management strategies of the PWUD unfold, embedded in a system of governance, which also limits the ability of the PWUD to shake the social space constructed by the management approach.

The absence of rehabilitation centers in the treatment management approach allows multiple identities to act in an unintegrated manner in the lives of PWUD rather than assisting them through sobriety and re-citizenship in a homogeneous, rehabilitative capacity. On their side, PWUD expect to return to a “non-surveillance, low-discrimination, dignified” life as ordinary citizens and thus has high hopes for the civic programmatic nature promised by the management approach. The logic of “citizenship” has been used to develop pathways and ways to become ordinary citizens again. This game of individual identity and political rights presents a serious social problem for drug users in the real world that goes far beyond the addictive nature of medical discourse.

As a result, a summary of the self-management strategies adopted by drug addicts reveals that these behaviors exhibit a more characteristic “disengagement model” than the established mainstream integration model, which is closely related to the Chinese context.

Self-management strategies and the self-narrative of “possible citizenship” cannot truly change the plight and fight for equal rights, even if used intentionally or unintentionally in their lives, due to their limited cognitive and social resources. PWUD’s self-management is embedded in the drug disciplinary system paradoxically, manifesting as a mode of disengagement alleviating the identity issue.

Identity conflicts, especially those of marginalized groups, have always been an issue of academic concern. In addition to analyzing the causes of conflicts, how to improve identity conflicts and avoid discrimination and injustice caused by identity fragmentation are also important research directions. 

In researching the LGBT+ community, Orit Avishai presents and discusses identity conflict, management, negotiation, and reconciliation in the LGBT+ community [46]. An extended way of understanding and de-escalating conflicting identities are offered—categorizing and integrating identity conflicts within a conflict framework. Rodriquez and Ouellette hold the same view, distinguishing between segregation and identity integration. In this model, identity integration is the most desirable outcome because it alleviates conflict and integrates religious beliefs and sexual identities into a single positive identity [47]. In turn, an analytical framework with identity conflict as a starting point and identity integration as an implicit goal was developed [48,49,50]. In general, Avishai’s understanding of identity conflict is context specific. Yet, there is a universal interpretation of identity contradictions and flows. It is instructive to look beyond “conflict”, for example, by focusing on the role of shame, loss, and social justice consciousness [51,52,53,54]. Such thinking about the classification, management, and integration of identity conflicts, with the goal of homogenization and first nature, can be called the “integration model”.

The (identity) integration model assumes that a unified positive identity will eventually replace conflicting identities, implying both the ability to reproduce conflicting identities and emphasizing the substitutability of conflicting identities, arguing that a consistent positive identity is a suitable substitute. The integration model also applies to the drug rehabilitation process, with Dingell focusing on the social identity pathways associated with drug and alcohol addiction and abstinence. Twenty-one adults living in drug and alcohol treatment communities were interviewed, and one group was found to believe that addiction was associated with some positive identity, viewing addiction as a loss of positive identities, and wanting to restore their pre-addiction social identities (renewed identities) through treatment. The other group sees addiction as a new and valuable gain of identity with addiction (gain of identity with addiction), and after treatment, they are more inclined to acquire new and more challenging roles (aspirational identities), including school, work, or family roles (Figure 2 depicts this process). Additionally, Dingle hit the nail on the head when he stated that the basis for leaving rehab and achieving both post-recovery identities is the establishment of the same recovery identity in a rehabilitation center (TC) that moderates the negative identity developed in addiction and the negative effects of the drug user identity [41].

However, specifically in the case of urban Chinese PWUD, they adopt a different type of action strategy, although they also perceive multiple identities and identity conflicts. Unlike Avicha’s conflict framework, which is more in line with Deleuze’s view of difference [55], they tend to use segregation as the primary strategy and do not desire and objectively lack structural organizations (rehabilitation centers) to achieve identity integration. Thus, Chinese PWUD alleviate their identity dilemma by not forming a single active identity but by forming diffuse and unfocused ordinary citizenship identities that assist them in being less monitored and politicized.

This disengagement model manifests itself in how PWUD use their limited resources to strip or conceal the negative aspects of their identity and repackage themselves as ordinary citizens. Disengagement thus points to a degree of separation (isolation) from the “drug use” and “stigmatized” identity symbols. Combining the previous analysis of management approaches and action strategies, the process of the disengagement model is mapped in Figure 3.

In contrast to the integration model, the disengagement model is not about the organization of recovery, such as rehabilitation centers or help groups, but about the PWUD themselves. Because of this quality, the disengaged model does not emphasize how actively they use social resources and reconstruct a de-stigmatized identity. The integrated model is more focused on this, as the addict chooses to use their resources to weaken real-life stigmas that point to them (e.g., Feng says (we) are not aggressive, have a good personality, and are willing to help) and hide their own “drug-using” label in social and work situations (e.g., Min says (co-workers) don’t know about it and I don’t talk to them, or I won’t be friends), and hiding their “drug use” label in social and work situations. In addition, PWUD are more focused on social experiences and power gains in the present and hold ambiguous judgments and representations of past experiences. They rarely envision how to achieve a future rehabilitative identity or a new positive social identity. Moreover, the futuristic nature of the integration model is more pronounced, emphasizing the reconfiguration of positive/new identities. However, PWUD prefer to avoid sticking to the past and to maintain and sustain the current, half-concealed, established effective socialization.

The disengagement model helps to reduce the frequency with which stigmatized identities and “drug-using” labels are emphasized in daily life and, to some extent, avoid the perpetuation and reiteration of past experiences. When dealing with the management of the CAG, communities, and police stations, some autonomy and self-awareness can be retained. Daily surveillance and law enforcement control can be weakened; in everyday life, detoxification can be achieved using “anonymous mobility (spatial isolation)”, “disconnection from past experiences (temporal isolation)”, and “instrumental strategies”, and the narrative of “possible citizenship”, the detachment model expresses the meaning of being able to help detainees continuously explore the possibility of re-establishing a dignified, independent life.

However, the disengagement model is always a path of civic program exploration that PWUD try to develop after perceiving a life dilemma. Two qualities cause it to degenerate into self-indulgence quickly. Periodic exposure from the help team and police station will often interrupt the disengagement model (how it is interrupted is mentioned in the section on identity management effectiveness), which may help to understand why detainees often do not cooperate when notified by drug workers about urine tests, even though they are not involved in relapse. Second, in a relatively closed disengagement model, most detainees are limited to all their resources, while others maintain relationship resources. In the face of management and law enforcement, the power of detainees to use their resources to resist and fight to preserve disengagement is weak. They often choose to compromise after a short period of stalemate, return to supervision and control, and wait for an opportunity to enter disengagement mode again.

## 7. Discussion: Can Self-Management Reduce the Stress from Policing and Management?

To cope with the dilemma of identity conflict, PWUD try to use hidden mobility (spatial isolation), complete rupture (time separation), and instrumental strategies to make themselves more like ordinary citizens, hide their personal experience of using drugs, and better integrate into society. We can define this process as “A quasi-citizen model developed by individuals”. Its primary purpose is to temporarily replace drug policies and legal assumptions that agents have not implemented. Interestingly, the PWUD interviewed did not fully understand the relevant laws and policies. Their sense of identity conflict is mainly derived from the police and the CAG’s requirements for control, surveillance, and differentiation from ordinary citizens. Although addicts do not understand the various factors of their difficulties, this does not affect the admiration of addicts for non-stigmatized identities and mainstream groups.

Therefore, can these identity management strategies help them break through the predicament and fight for more rights? The answer is no. 

From an empirical view, Chinese PWUD conduct and manage their identity through an ambiguous relationship with disciplinary regimes. In the spatial isolation strategy, PWUD place themselves in a new, unfamiliar relationship and conceal the CAG trajectory to reshape their new personal identity and prevent exposure to their drug-using experience. However, the CAG and the police will immediately believe that PWUD have relapsed without repeated inspections and interviews. They will require the PWUD to participate in a urine test through threats and warnings. Therefore, this disengagement mode is only temporary and highly risky. Broken, the strategy of covering up past experiences in the dust is, even more, a measure of self-deception. If community drug rehabilitation is not completed, past drug use will always be aroused in interviews. Even PWUD who have completed detoxification and returned to society will be suddenly awakened from memory, which is like a nightmare constantly repeating. Instrumental cooperation is one of the most successful methods to benefit subsistence allowance, medical insurance, and housing. However, when survival needs are met, PWUD desire a decent life with dignity. Then, they will lose because of a hopeless lifestyle. For example, they hope for welfare jobs, opportunities for re-education, or good interpersonal relationships, but this often involves the values and ideologies of the entire society on drug use, which is difficult to achieve. If the initial instrumental cooperation obtains resources through a stigmatized identity to survive, the later instrumental collaboration cannot get better resources.

In addition, self-management strategies also exhibit three essential characteristics: highly normative, avoiding conflict, and single resources, which leads to limitations and ineffectiveness. Highly normative is that PWUD often express the desire to return to the mainstream culture. That is, in addition to their past using drug experience, they are more supportive of the government and the country’s big decisions than others. Avoiding conflict is trying to escape or cut into the relationship with the original community, peers, and self, aiming to create a stable and private daily life outside the shadow of legal control. Single resources are the resources to support the strategy implementation comes from the life of the PWUD, lack of sustainability and external resources, and are prone to interruption and duplication.

These three characteristics of self-management indicate that the strategies of PWUD to cope with identity issues are compliant with mainstream culture, limited to their resources, and have not formed a sub-cultural practice that confronts management power and seeks to fight for their rights from the bottom up. Therefore, the self-management of PWUD is only a limited self-protective passive action under the disciplinary system.

The self-management strategy is an attempt to depoliticize. As the objects of the governance system, PWUD also are aware of the ideologies of national security, social citizenship education, and public risk, but they differ from the sub-cultural resistance mode. They strive to maintain consistency with the country and society, showing that they are citizens who love their country, the party, and society and hiding their identity and experience of PWUD from those around them. This unanimity of imagination enables the addicts to gain part of their life autonomy.

## 8. Conclusions 

In implementing drug treatment management, the executive agencies represented by the NCO, the police station, and CAG have formed a diverse governance model with different working concepts. This governance has caused multiple and incomplete identity dilemmas for PWUD. As a result, PWUD have been living a surveillance life of stigmatization, high-level control, and identity separation for a long time to alleviate the discomfort of life. Urban PWUD have adopted strategies such as hidden mobility (spatial isolation), complete rupture (time separation), and instrumental cooperation is adopted to hide personal experiences of using the drug and try to return to ordinary citizenship. However, in this game of individual identity and political rights, the strategies of PWUD have not changed the essence of “Incomplete Citizenship”. Their behavior model, the disengagement model, presents this process.

The contribution of this article is reflected in the following three points. Firstly, in the part of empirical research on the daily life of addicts, it provides findings that are different from similar studies in the United States. Existing research on the identity conflicts of marginalized groups often summarizes the actions of marginalized groups as sub-cultural movements with resistance characteristics. However, Chinese urban PWUD use self-management strategies to adjust to the social pressure and injustice they feel and actively move closer to the mainstream culture. The marginal and mainstream groups reach a state of symbiosis rather than confrontation under the same norms. 

Then, at the theoretical level, we summarize and reflect on the following two points: First, around citizenship, we present in detail the underlying paths to citizenship for marginalized groups (PWUD, for example), which are internally segmented from a cultural perspective, dividing citizenship into “incomplete citizenship,” “ possible citizenship,” and “ordinary citizenship.” To obtain the same “ordinary citizenship” as ordinary people, PWUD perceives the characteristics of its “incomplete citizenship” and experiences it as “possible citizenship” through self-management strategies. To obtain the same “ordinary citizenship” as ordinary people, PWUD experienced the political life of “ordinary citizenship” through self-management strategies after recognizing the characteristics of their “incomplete citizenship.”

Second, to summarize the anti-drug practice model, PWUD adopt a disengagement model that is different from the integration model because of the lack of a social structure space for integrated rehabilitation and social rehabilitation through themselves. As a result, PWUD in China have been unable to create a new valued identity that bridges the addiction experience and have instead worked to return to a pre-experience, non-socially excluded citizenship. This disengagement model presents conservative values due to its low creativity and regressive feature.

The identification of barriers to PWUD’s social integration has meaningful implications for rehabilitation success in the broader drug governance system. Identity conflicts and failure of self-management are not unique to rehabilitation projects at the street level, prior literature has identified similar dynamics of exclusion in other agencies of drug control and treatment [41]. If barriers in the path from rehabilitation to social integration are consistent across multiple settings, it may benefit practitioners and policymakers to coordinate their efforts and launch initiatives that targe the overall institutional logic rather than the specific techniques. For the drug control system, it means to change the organizational structure of the existing anti-drug system and establish voluntary rehabilitation centers coordinated by the market and the government, which allow more free choices on diverse drug treatment services and less top-down monitoring and political pressure. For the rehabilitation treatment professionals, it means helping PWUD develop an identity integration model or a unified positive identity that will eventually replace conflicting identities. The integration model will facilitate PWUD’s access to available social resources and intentionally construct institutional scenarios for de-stigmatization to break down the segregation between PWUD and other social groups.

Of course, we must acknowledge this study’s limitations. First, the study was conducted in city X only, and similar studies need to be conducted in other cities in China in the future to generalize the disengagement model of self-management. Second, taking an “environment-behavior” perspective, we explain the behavioral pattern of PWUD mainly through the lens of institutional environmental effects that consciously set aside the heterogeneity of psychological status and ideational scripts between individual PWUD. Further research on the psychological dimension of identity conflicts and their effect on PWUD’s health and social life should be put on the agenda.

PWUD’s self-management strategies in legal and political systems draw attention to how people with identities such as the governed, the marginalized, and the legal fringe survive and explore their lives in a compressed political space. Their actions and voices are also valuable in helping the judiciary reconstruct drug laws and institutions and assisting social workers and therapists in practicing empathy work.

## Figures and Tables

**Figure 1 behavsci-12-00258-f001:**
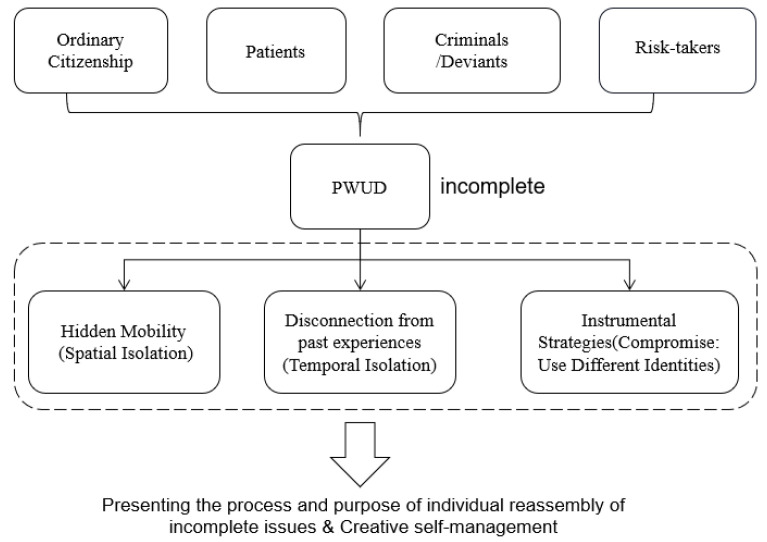
Incomplete issues and creative self-management of PWUD.

**Figure 2 behavsci-12-00258-f002:**
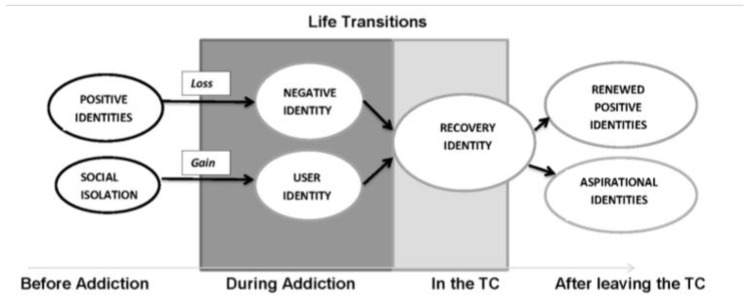
Diagram of page 5 of Dingle’s “Social Identities as Pathways”.

**Figure 3 behavsci-12-00258-f003:**
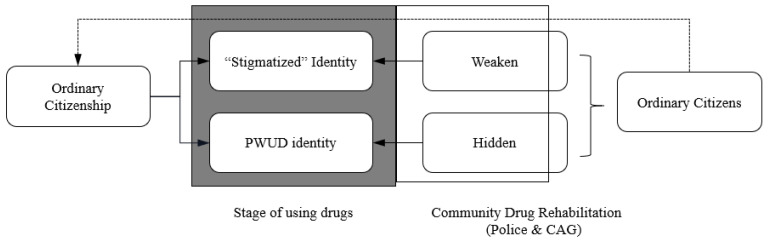
Processes of disengagement model among urban PWUD in China (compare to Dingle).

**Table 1 behavsci-12-00258-t001:** Information of in-depth Interviewee.

NO.	Nickname	Age	Gender	Education (Years ^1^)	Role	Drug Using History (Years)	Work or Not
A	Feng	50	Male	6	PWUD	10	No
B	Xin	60	Male	6	PWUD	20	No
C	Song	43	Male	16	PWUD	5	Staff in the market
D	Ping	33	Female	8	PWUD	21	No
E	Min	41	Male	9	PWUD	12	delivery man
F	Wen	31	Female	6	PWUD	12	No
G	Zhong	48	Female	9	PWUD	15	No
H	Huang	26	Female	16	Staff of CAG	None	Social worker
I	Lee	35	Male	16	Leader of NCO	None	Capitan
J	PH	37	Male	13	policeman	None	policeman

^1^ Education years start from primary school.

**Table 2 behavsci-12-00258-t002:** Multiple governance and identity dilemma.

Governance Subject	Specific Content	Identity	Dilemma
Drug Law and Policy	The Drug Law and the Regulations.	Quasi-ordinary citizens	multiple and incomplete identities
National Agency	Narcotics Control Office (NCO) Police Station Community Assistance Group (CAG)	Patients	
		Deviants or Criminals	
		Risk-takers	

## Data Availability

The materials that support the findings of this study are openly available in Open Science Framework’s online depository, available at https://osf.io/658dv/ (accessed on 14 May 2022). Data were collected and uploaded by the author of this article.

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
