# Peer review of "“Returning to Ordinary Citizenship”: A Qualitative Study of Chinese PWUD’s Self-Management Strategies and Disengagement Model of Identity"

_behavsci, 2022, doi:10.3390/bs12080258_

Round 1
Reviewer 1 Report
Thank you for giving me the opportunity to review this article. Please see my comments below to further improve it.
Certain statements require clarity. For eg.
Line 55 -57: In other words, the lack of 55 description and analysis of the survival situation and living conditions of PWUD in China 56 is no way to ensure that the critiques and recommendations of established research on a 57 state agency, family, and community are justified?
The research question given in the intro is ambiguous. Authors need to rephrase the same.
More information is required on the in depth interviews conducted by the authors. The procedure given in the paper doesn't give any information on the pattern of the interviews, how they respondents were probed, structure and type of questions used and so on.
The authors need to work on the coherence and flow of the themes that they have added in the paper. The readers may find it difficult to connect the ideas with the current structure.
The policy implications of the study could be added as a separate sub section.
Author Response
Dear ME Li and referees,
Thank you very much for your works on this manuscript, I have read comments carefully and responded point by point to all of them. The reviewers' suggestions on details of the article such as formatting, citations, abbreviations, and constructive comments on the methodology, conclusion, and introduction sections of the article were very helpful in enhancing this paper to be more relevant to the theme of the journal as well as to better meet the needs of the readers. Once again, we thank again the three reviewers for their review. Below we present the changes of the article after incorporating the reviewers' comments.
The response to Reviewer 1 is as follows:
1.the mistaken expression of line 55 -57.
Response: We correct these two lines of clerical errors and rephrase line 55-57.
2.Research question should be spelled out more clearly in introduction.
Response: On page 2, we clarify and highlight our research questions to make it more outlined in introduction.
Detailed revision: line 63-74
This study illustrates how and why Chinese PWUDs use self-management strategies to cope with the identity issues of drug governance and incomplete citizenship within institutional arrangements characterized with social exclusion. Empirically, this study focuses on PWUDs’ daily life, specifically, the experiences and practices of de-stigmatizing life in the context of social control, which is the core of the study from a cultural and life-course perspective. The first part of the analysis describes PWUDs’ lived experience of the identity conflicts and their perception of incomplete citizenship in everyday life situation of drug governance. The second part examines their strategies of self-management to alleviate the identity conflicts and conceptualizes into the model of disengagement. Finally, we discuss the extent to which self-management strategies and self-narratives of "possible citizenship" help them break through the identity predicament, gain life autonomy, and enhance social integration.
3.More information on procedure of in-depth interviews should be provided.
Response: On page 5, we added a new paragraph to the sub-section (3.1) to illustrate selection process of interviewees. That paragraph provides details about how our cooperation with Social Organization Q helped us access to PWUDs’ profiles, contact with eligible interviewees, and get acquainted with working staffs of rehabilitation projects.
On page 5, we added a new paragraph to the sub-section (3.2) to elucidate the outline of interview questions. Four mutually connected themes structured the interview outline: perception and response to stigma, interpretation and justification of drug use, drug governance policy and control agents, and strategies to gain social integration. This paragraph also provides more information on our selection of interviewees.
On page 6 and 7, we inserted a new sub-section (3.3) “Data processing and coding” to enrich the methodology of this research. This sub-section introduces how interview data is transcribed and verified and coded. We also provide Appendix Table A1, A2, and A3 on page 21-23 to introduce this data processing and coding process in detail. Table A1 presents the transcription and verification of interview materials. Table A2 provides information on the principles and procedures of open and spindle coding. Table A3 shows the deeper stage of data recoding as generic analysis of "context-restriction-behavior-result".
4.Strengthen the coherence of the themes to make it fit the structure better.
Response: In introduction, we added a brief introduction of the following structure (on page 2), so that readers will have a guideline of themes. And we have also tried to make a top-down statement before and after the chapters, within the subsections, and in the concluding section, we have again made a summary and structural statement of the whole text.
The first part of the analysis describes PWUDs’ lived experience of the identity conflicts and their perception of incomplete citizenship in everyday life situation of drug governance. The second part examines their strategies of self-management to alleviate the identity conflicts and conceptualizes into the model of disengagement. Finally, we discuss the extent to which self-management strategies and self-narratives of "possible citizenship" help them break through the identity predicament, gain life autonomy and enhance social integration.
5.A separate sub-section of policy implications should be added.
Response: On page 20, we write a new paragraph to illuminate policy implications of our findings. In this paragraph, we emphasize the importance of identifying barriers to PWUD’s social integration and the necessity of developing voluntary rehabilitation center as well as cultivating an integration model of PWUD identity. Details is here:
The identification of barriers to PWUD's social integration has meaningful implications for rehabilitation success in the broader drug governance system. Identity conflicts and failure of self-management are not unique to rehabilitation projects at street level, prior literature has identified similar dynamics of exclusion in other agencies of drug control and treatment [41]. If barriers in the path from rehabilitation to social integration are consistent across multiple settings, it may benefit practitioners and policy makers to coordinate their efforts and launch initiatives that targe the overall institutional logics rather than the specific techniques. For the drug control system, it means to change the organizational structure of existing anti-drug system and establish voluntary rehabilitation centers coordinated by the market and the government, which allow more free choices on diverse drug treatment service and less top-down monitoring political pressure. For the rehabilitation treatment professionals, it means to help PWUDs develop an identity integration model or a unified positive identity that will eventually replace conflicting identities. The integration model will facilitate PWUDs access available social resources and intentionally construct institutional scenarios for de-stigmatization to break down the segregation between PWUDs and other social groups.
Of course, we must acknowledge this study’s limitations. First, the study was conducted in city X only, and similar studies need to be conducted in other cities in China in the future to generalize the disengagement model of self-management. Second, taken an "environment-behavior" perspective, we explain the behavioral pattern of PWUDs mainly through the lens of institutional environmental effect that consciously set aside the heterogeneity of psychological status and ideational scripts between individual PWUDs. Further research on the psychological dimension of identity conflicts and its effect on PWUDs’ health and social life should be put on the agenda.
Thanks again for your review.
Warm regards,
Song and Liu

Reviewer 2 Report
Thank you very much for this opportunity to revise the manuscript titled "Returning to Ordinary Citizenship": Chinese PWUD's Self- 2
Management Strategies, Purpose, and Model " that was submitted to Behavioral Sciences (Special Issue - Clinical Psychology Research and Public Health).
First of all, congratulations for choosing a highly topical issue.
Additionally, I leave several comments about the manuscript which are listed below:
1. Results of abstract should be more informative and very important findings should be mentioned, briefly.
2 First and second paragraphs (2.1. and 2.2.) of Literature reviewshould be integrated and, extremely, summarized.
3. Method and Analysis framework were described in details and both sections are too long. I recommend summarizing these sections by authors if it`s possible.
4. Limitations and Suggestions for future studies as well as practical implications should be added, substantially.
This paper holds potential value to the readers on Behavioral Sciences.
I would be glad to see this article in the second round.
Author Response
Dear ME Li and referees,
Thank you very much for your works on this manuscript, I have read comments carefully and responded point by point to all of them. The reviewers' suggestions on details of the article such as formatting, citations, abbreviations, and constructive comments on the methodology, conclusion, and introduction sections of the article were very helpful in enhancing this paper to be more relevant to the theme of the journal as well as to better meet the needs of the readers. Once again, we thank again the three reviewers for their review. Below we present the changes of the article after incorporating the reviewers' comments.
The response to Reviewer 2 is as follows:
1.Findings should be mentioned more briefly in abstract.
Response: Thanks for your comments. We revised abstract and added some findings in it on page 1. Detail is here:
How PWUDs (people who use drugs) live under drug governance is an important research question. Using a qualitative research method, this study explores how PWUDs in China self-manage after perceiving the dilemma of incomplete citizenship governance and analyzes the behavioral characteristics of self-management strategies.. The study found that PWUDs adopted behavioral strategies such as hidden mobility (spatial isolation), disconnection of past experiences (time isolation), instrumental actions and attempt to change incomplete citizenship by describing themselves as ordinary citizens in terms of possible citizenship narratives. Further, PWUD's self-management strategies manifest as a disengagement model in which the actors (PWUDs, not rehabilitation agencies) do not work to build integrative positive identities through dispersed, practiced behavioral strategies, but rather attempt to return to pre-addiction, non-socially exclusionary citizenship experiences. Ultimately, the disengagement model fails to create effective political influence on policy and management because of its single, non-resistant resource, but PWUD's actions and voices from the bottom can still help us reflect on current drug governance and social work.
2.Literature review 2.1. and 2.2. should be integrated and summarized.
Response: We have reorganized the content of review sections 2.1 and 2.2 to consolidate and further refine this section due to its contiguous relationships and overlapping content. The revised literature review section consists of only one section. We have integrated the overview of identity shaping into the ideologically guided policy and organizational practice section of the harm reduction. In addition, we have reduced the section listing specific experiences and refined and integrated their content. The specific overview sections are presented as:
There are mainly two types of issues on PWUD: how to achieve rehabilitation and how to reduce harm. The former focuses on the recovery of physical health to interpret the physical and emotional changes of PWUD through the mechanism of SUD (substance use disorder) and addiction [7—12]. The latter has a broader scope of discussion, following with rehabilitation treatment, primary physical health injuries, justice and recovery in social relations, political situation, and rights of PWUD [4—6]. Therefore, studies on harm reduction must pay attention to the influence of authority, family structure, and social structure on PWUD. It presents a top-down analysis context and relies on a politicized approach.
Using strategies to reduce harm is filled with the imagination of the governing subjects. In practicing these ideologies, some help the rehabilitation of PWUD, and some invisibly aggravate their harm, including social discrimination, damage to self-esteem, and leading an unhealthy life [13-14]. Lynda believes that understanding the drug problem requires value neutrality [15]. The government and service departments should be responsible for their remarks to guide the implementation of treatment as soon as possible.
The ideology of harm reduction guides the research and practice of social identity rehabilitation of PWUDs to assist them in achieving physical rehabilitation and reconstructing their identity in society. These studies on reconstructing the rehabilitation social identity of PWUDs are mainly structured [16—19], using structural power to assist PWUD in achieving rehabilitation. Structural power presupposes the type and situation of PWUD and sets up what they should look like after successful rehabilitation. It is hoped that addicts will no longer be shackled by addiction in the future. In his discussion of the battle for Chastity, Foucault sees Gesian’s steps in outlining the struggle for Chastity as a process of continuous abstinence, intending to allow the subject to achieve “unmoved by evil”. Reconstructing the identity also aims at rehabilitation and abstinence, expecting that the addict will not be shackled by addiction in the future. Its operation can be broadly summarized into two categories: structural and critical shaping.
Structural shaping refers to constructing normative identity acquisition pathways through structural forces, mainly including “state—citizen” and “social network—citizen.” For example, drug research in Australia has focused on how to solve drug use problems and effectively change the situation of PWUD from the perspective of law and drug policy, tensions, and dilemmas [1] [16—17] [20]. However, this shaping, with arrest, incarceration, and reform through labor as actions, reinforces PWUD’s fear of criminal justice, and although detention is always in the name of detoxification, it creates persistent occupational hazards and integration dilemmas in practice [21—22]. Therefore, PWUD either uses Alice Goffman’s proposed escape route to evade capture or actively resists criminal justice to build the moral system of the incarcerated community in the form of criminal “honor” [18] [23]. In addition to policy and law, governmentality also plays a vital role. Beth proposes neoliberal governance techniques that focus on how governments intervene with populations to (re)produce ideal citizens. For example, the development of biopolitics focuses on a biological perspective on population, with surveillance and individual responsibility pressures as strategies to preserve population health. Thus, they emphasize the critical relationship and combination between biological identity. And The state, the market, and biomedicine focus on reparations for collective personal harm as the possible governance models for biopolitics. Scholars such as Ingrid discuss how public health strategies can reshape forms of autonomy and citizenship for people living with HIV, and studies have found that exemplary citizenship for stigmatized groups is difficult to follow through and sustain because of the complexity and inequality [24].
The “social network—citizen” manifests itself as a process of shaping PWUD by relational forces. Family and community are among the essential spaces for identity remodeling and social identity reconstruction, assisting PWUD in transitioning from social networks of abuse to recovery-supportive social networks [25]. Especially the process of identity change within recovery is essential, through which new identities replace stigma and addict identity, and recovery capital and quality of life are improved [19].
Critical shaping refers to the critique of the established set of criteria that define drug users as a stigmatized, marginalized, and pathologized group, manifested in a critique of medical and biological discourses and an emphasis on neoliberal self-management. In classical studies, PWUD’s bodies are constantly reduced to tissues, blood, and neural pathways as a de-subjectified generation of enjoyment or pain [26]. To argue and understand how the body is manipulated by the “substance” and whether it is accepted or feared in response to the manipulation, thus constructing a negative association between certain addictive behaviors, especially drug use, and physical, psychological, and neuronal [27—29].
Medical discourse has effectively promoted the healthy expression of the drug phenomenon and the pathologization of PWUD [30]. While the eugenic philosoph of biological development in the first half of the 20th century gradually popularized the conceptual model of non-defective joint disease, making PWUD biologically judged as vulnerable, defective, and dangerous individuals. Thus, it critiqued the medical and biological model as an essential research direction since the late 20th century [31]. Two of these critical assumptions are relatively well established,
The first promotes de-pathologization and de-biological differentiation, thinking about drug addiction on a relational level [32] and from a cultural perspective [26]; the second is eclectic, using a balanced or integrated strategy outside of material determinism (neuroscience) and cultural influences [33—34], focusing on the narratives of drug users and the complexity and diversity of PWUD’s actions in social settings [35—36].In narrative studies of PWUD, positive risk-taking and self-management characteristics are evident. For example, in the Swiss “recreational” using narrative, PWUD reflect on the blurred division between “safe” and “dangerous” and advocate for a change in the old judgments of “drug use” in the addiction recovery space to re-examine the association between drug users and drug use behaviors [35].
There is a clear difference between structural and critical shaping: the former takes the medical-biological discourse as the basis, through the power of the state and the power of relationships, to enable drug users to reach a standard of “health” in a normative and legal path. The latter critiques medical and biological discourses as being too homogeneous in their means, advocate attention to PWUDs’ plurality and complexity in social contexts by reconstructing authoritative discourses and balancing cultural-biological positions and emphasizes understanding PWUD’s narratives to help them develop self-management skills. The significant commonality between these two shaping approaches is that they both aspire to adjust the relationship between the disordered “individual” and society so that they can live without discrimination and exclusion.
Some problems cannot be realized in transforming the value of harm reduction from ideology to practical operation. Analyzing this gap, we need to pay attention to how the rehabilitation process of PWUDs work in specific situations. The real problems are different from the original intention of reducing harm. The initial purpose of constructing identity is perhaps to reduce harm, but in the practice of management, if constructing ideas that ignore the opinions of PWUD, this political system creates all kinds of contradictions in grassroots practice. Therefore, it is necessary to carefully examine the understanding and actions of PWUDs, take the actions to deal with governance and difficulties, and reflect on the limitations of political imagination in governance.
3.The section of Method and Analysis Framework needs to be shortened.
Response: Thanks to your suggestion, we have trimmed this section. In (3.1) Field Situation, after describing the methodology used, we trimmed down the detailed description of the literature research methods and instead briefly explained how the materials were obtained and how the interviewees were approached. In (3.2) Procedure, we trimmed down the detailed interview content and moved it to the appendix and supplementary sections, including how the materials were handled and the outline of the interview. The revision are presented as follows:
Social Organization Q is the gatekeeper and main introducer of our fieldwork, through which we developed connections with local police station, CAG (Community Assistance Group), and three communities willing to introduce their drug rehabilitation projects. In addition, we selected PWUD interviewees from Q’s registration system that covered drug rehabilitation project sites across five blocks and recorded more than 130 biographical profiles of registered PWUDs (males are 66, around 73%, and females are 24). We asked Q to give referrals at the beginning of fieldwork and snowballed the following interviewees through them to provide other survey objects belonging to the overall research target and selecting the following survey objects according to the former’s recommendation.
......
The in-depth interview was divided into two parts: interview with PWUD and with Staff. The Interviewing with PWUDs focused on the following aspects: perception and response to stigma, understanding and narrative of drug use, health significance, national drug control agents, strategies, and actions integrated into life (supplementary materials). Then talking with the NCO and CAG in four aspects: management methods and requirements, how to carry out the agency’s tasks, how to distinguish and judge PWUD’s rehab status, and how to interact with them in the agency. After finishing the preliminary interviews, 14 PWUDs and 10 staff were interviewed. We stopped here because ten depth-interviewees have produced saturation. Then, we focused on the follow-up interviews with 10 participants, whose narratives constitute the core points of this article.
.......
4.Limitations and Suggestions for future studies as well as policy implications should be added.
Response: Thanks so much for your comments. These parts are very important for us to rethink our research and improve this manuscript. We added limitations, suggestions for future studies and policy implications in the conclusion section. The revision are presented as follows:
The identification of barriers to PWUD's social integration has meaningful implications for rehabilitation success in the broader drug governance system. Identity conflicts and failure of self-management are not unique to rehabilitation projects at street level, prior literature has identified similar dynamics of exclusion in other agencies of drug control and treatment [41]. If barriers in the path from rehabilitation to social integration are consistent across multiple settings, it may benefit practitioners and policy makers to coordinate their efforts and launch initiatives that targe the overall institutional logics rather than the specific techniques. For the drug control system, it means to change the organizational structure of existing anti-drug system and establish voluntary rehabilitation centers coordinated by the market and the government, which allow more free choices on diverse drug treatment service and less top-down monitoring political pressure. For the rehabilitation treatment professionals, it means to help PWUDs develop an identity integration model or a unified positive identity that will eventually replace conflicting identities. The integration model will facilitate PWUDs access available social resources and intentionally construct institutional scenarios for de-stigmatization to break down the segregation between PWUDs and other social groups.
Of course, we must acknowledge this study’s limitations. First, the study was conducted in city X only, and similar studies need to be conducted in other cities in China in the future to generalize the disengagement model of self-management. Second, taken an "environment-behavior" perspective, we explain the behavioral pattern of PWUDs mainly through the lens of institutional environmental effect that consciously set aside the heterogeneity of psychological status and ideational scripts between individual PWUDs. Further research on the psychological dimension of identity conflicts and its effect on PWUDs’ health and social life should be put on the agenda.

Reviewer 3 Report
Compelling and interesting article. I do believe there are some things that will make it stronger after revision:
1. I think the purpose of the manuscript needs to be made more explicit and clearer in the introduction.
2. Stronger (more developed) conclusion. The general points are solid, but I believe you need to more clearly connect your data/findings to the key takeaways empirically.
3. Need to provide a clearer and more defined section on methodology (how participants selected, interview questions, etc.)
4. Other less important but still relevant issues:
PWUD should be spelled out on first usage (even with the abbreviation table). SUD also should be spelled out on first usage. Please make sure this is done for all abbreviations as the journal has a broad audience.
Some formatting and style issues (punctuation, etc.) to address.
Citations of authors in some cases use first name (Alice, Renee, etc.) instead of last name.
Not clear why material is italicized beginning on line 148 (and in other places as well).
Reference list is not arranged alphabetically in all places. Also, please make sure all references are formatted completely.
Author Response
Dear ME Li and referees,
Thank you very much for your works on this manuscript, I have read comments carefully and responded point by point to all of them. The reviewers' suggestions on details of the article such as formatting, citations, abbreviations, and constructive comments on the methodology, conclusion, and introduction sections of the article were very helpful in enhancing this paper to be more relevant to the theme of the journal as well as to better meet the needs of the readers. Once again, we thank again the three reviewers for their review. Below we present the changes of the article after incorporating the reviewers' comments.
The response to Reviewer 3 is as follows:
1-Research purpose should be spelled out more explicit in the introduction.
Response: Thanks for your very important comment. The purpose of the study and the question should be mentioned and clearly explained in the introduction section, and we strongly endorse this suggestion. Therefore, we have added a paragraph to clarify and explain them. It is:
This study illustrates how and why Chinese PWUDs use self-management strategies to cope with the identity issues of drug governance and incomplete citizenship within institutional arrangements characterized with social exclusion. Empirically, this study focuses on PWUDs’ daily life, specifically, the experiences and practices of de-stigmatizing life in the context of social control, which is the core of the study from a cultural and life-course perspective. The first part of the analysis describes PWUDs’ lived experience of the identity conflicts and their perception of incomplete citizenship in everyday life situation of drug governance. The second part examines their strategies of self-management to alleviate the identity conflicts and conceptualizes into the model of disengagement. Finally, we discuss the extent to which self-management strategies and self-narratives of "possible citizenship" help them break through the identity predicament, gain life autonomy, and enhance social integration.
2-Findings should be connected to the key points empirically more closely in the section of conclusion.
Response: We agree with this comment. In the initial manuscript, our main work in the conclusion section was to summarize the findings and present the contribution points, and after the revision we added “Limitations and Suggestions for future studies as well as policy implications” to provide a closer, practical and concrete reflection on key points. Details is here:
The identification of barriers to PWUD's social integration has meaningful implications for rehabilitation success in the broader drug governance system. Identity conflicts and failure of self-management are not unique to rehabilitation projects at street level, prior literature has identified similar dynamics of exclusion in other agencies of drug control and treatment [41]. If barriers in the path from rehabilitation to social integration are consistent across multiple settings, it may benefit practitioners and policy makers to coordinate their efforts and launch initiatives that targe the overall institutional logics rather than the specific techniques. For the drug control system, it means to change the organizational structure of existing anti-drug system and establish voluntary rehabilitation centers coordinated by the market and the government, which allow more free choices on diverse drug treatment service and less top-down monitoring political pressure. For the rehabilitation treatment professionals, it means to help PWUDs develop an identity integration model or a unified positive identity that will eventually replace conflicting identities. The integration model will facilitate PWUDs access available social resources and intentionally construct institutional scenarios for de-stigmatization to break down the segregation between PWUDs and other social groups.
Of course, we must acknowledge this study’s limitations. First, the study was conducted in city X only, and similar studies need to be conducted in other cities in China in the future to generalize the disengagement model of self-management. Second, taken an "environment-behavior" perspective, we explain the behavioral pattern of PWUDs mainly through the lens of institutional environmental effect that consciously set aside the heterogeneity of psychological status and ideational scripts between individual PWUDs. Further research on the psychological dimension of identity conflicts and its effect on PWUDs’ health and social life should be put on the agenda.
3-The paper needs to provide more details on methodology.
Response: On page 5, we added a new paragraph to the sub-section (3.1) to illustrate selection process of interviewees. That paragraph provides details about how our cooperation with Social Organization Q helped us access to PWUDs’ profiles, contact with eligible interviewees, and get acquainted with working staffs of rehabilitation projects.
On page 5, we added a new paragraph to the sub-section (3.2) to elucidate the outline of interview questions. Four mutually connected themes structured the interview outline: perception and response to stigma, interpretation and justification of drug use, drug governance policy and control agents, and strategies to gain social integration. This paragraph also provides more information on our selection of interviewees.
On page 6 and 7, we inserted a new sub-section (3.3) “Data processing and coding” to enrich the methodology of this research. This sub-section introduces how interview data is transcribed and verified and coded. We also provide Appendix Table A1, A2, and A3 on page 21-23 to introduce this data processing and coding process in detail. Table A1 presents the transcription and verification of interview materials. Table A2 provides information on the principles and procedures of open and spindle coding. Table A3 shows the deeper stage of data recoding as generic analysis of "context-restriction-behavior-result".
4-All abbreviations should be spelled out on first usage.
Response: This is a very important suggestion for specification and ease of reading, which we have adopted and revised. Details is here:
Line 9: How PWUDs (people who use drugs) live under drug governance is an important research question.
Line 78: ...through the mechanism of SUD (substance use disorder) and addiction [7—12].
Line 471: ...NCO (Narcotics Control Office), and three communities.
Line 475:...connections with local police station, CAG (Community Assistance Group),
Line 567:...a CDI (Compulsory Drug Institution) for drug-related crimes.
5-Specific points on addressing formatting and style issues.
Response: Thanks for your careful review and for your valuable suggestions in such detail, which we have accepted and revised, specifically:
Sp 1. We correct our citations and make the citations Meet the "behavioral sciences" requirement.
Sp 2. Adjusted article images, table formatting, and added appendix and supplemental information
Sp 3. We reformat our reference list to make sure it is arranged alphabetically.
Sp 4. The italicized content, which did not fit the format was removed. Only the cited interview material was kept in italics for the time being, which will be adjusted according to the editorial requirements.

Round 2
Reviewer 1 Report
The authors have addressed all my comments.
Author Response
Dear Reviewer 1,
Thanks for your review of this, Ms.
Bests,
Apei
Reviewer 3 Report
I appreciate the time and attention put into the revisions by the authors, including my comments as well as those of the other reviewers. I believe the manuscript is significantly improved.
Author Response
Dear Reviewer 3,
Thanks for your review. Yes, according to the referees' comments, this Ms has significantly improved.
Bests,
Apei